# Effectiveness of Abdominal and Gluteus Medius Training in Lumbo-Pelvic Stability and Adductor Strength in Female Soccer Players. A Randomized Controlled Study

**DOI:** 10.3390/ijerph18041528

**Published:** 2021-02-05

**Authors:** Héctor Guerrero-Tapia, Rodrigo Martín-Baeza, Rubén Cuesta-Barriuso

**Affiliations:** Department of Physiotherapy, Faculty of Sport Sciences, European University of Madrid, 28670 Madrid, Spain; hector.guerrero.t25@hotmail.com (H.G.-T.); rodrimb21@hotmail.com (R.M.-B.)

**Keywords:** core, isometric abdominal training, gluteus medius, lumbo-pelvic stability, physiotherapy

## Abstract

Background. Abdominal and lumbo-pelvic stability alterations may be the origin of lower limb injuries, such as adductor pathology in soccer players. Imbalance can be caused by both intrinsic and extrinsic factors. Methods: In this randomized controlled trial over 8 weeks, 25 female footballers were randomly allocated to an experimental group (isometric abdominal training and gluteus medius-specific training) or a control group (isometric abdominal training). Evaluations were performed at baseline, at the end of the intervention and after a 4-week follow-up period. The exercise protocol in common for both groups included three exercises: Plank, Lateral plank and Bird dog. Specific exercises for the gluteus medius were: Pelvic drop and Stabilization of the gluteus medius in knee valgus. Outcome measures were lumbar-pelvic stability and adductor strength. Results: After the intervention, there was an increase in lumbo-pelvic stability in both groups, being greater in the control group than in the experimental group (mean differences [MD]: 4.84 vs. MD: 9.58; *p* < 0.01) with differences in the analysis of repeated measures (*p* < 0.001), but not in group interaction (*p* = 0.26). Changes were found in adductor strength in the experimental group (MD: −2.48; *p* < 0.001 in the left adductor; MD: −1.48; *p* < 0.01 in right adductor) and control group (MD: −1.68; *p* < 0.001 in the left adductor; MD: −2.05; *p* < 0.001 in the right adductor) after the intervention, with differences in the analysis of repeated measures in left (*p* < 0.001) and right (*p* < 0.001) adductor strength. Conclusions: An abdominal and gluteal training protocol shows no advantage over a protocol of abdominal training alone for lumbo-pelvic stability and adductor strength, while improvements in both variables are maintained at four weeks follow-up.

## 1. Introduction

Soccer is a sport played by more than 260 million people all over the world, having various disciplines, including women’s soccer. This sport can generate structural and functional alterations, soft tissue injuries being especially common [1,2,3,4]. 

A number of etiological agents that may contribute to injury have been reported. In addition to the repetitive nature of the sports movement and the systematic overuse in soccer players, a poor physical condition which fails to meet the requirements for a given sport is highly relevant. A central core of stability for safe performance of the sporting movement is especially important, because poor stabilization can lead to injury, for example, of the adductors, this being very common in football [5,6,7].

Various methods of prevention and recovery for football injuries have been developed, but there is a clear trend in the design of such methods which are directed to men’s soccer [8,9,10]. Yet, anatomical features in males and females are not the same, and such methods must be approached in a relatively different manner [11].

Hölmich et al. [8] developed a program based on (concentric and eccentric) strengthening, coordination and stability exercises for the pelvic muscles, which attempts to address the prevention of pubic and inguinal injuries [10,12]. A number of authors have implemented this program over a 6-week period, demonstrating its efficacy in high-performance athletes to strengthen the muscles inserted in the pubic region [8,9,10,12]. However, such studies fail to show benefits for lumbar-pelvic and abdominal stability, something which, according to other authors, constitutes the basis of injury prevention in high-performance sports [3,6,7,13].

Structures such as the gluteus medius are essential to ensure proper stability. However, the specific training of muscles such as the gluteus medius has not been included in protocols described to date [9,13,14,15].

The purpose of this study was to establish if the inclusion of gluteus medius-specific exercises in an abdominal training program is more effective for improving lumbo-pelvic stability and adductor muscle strength, in semi-professional female soccer players aged from 18 to 35 years. Based on the above evidence stressing on the importance of stability training for injury prevention, a hypothesis was proposed whereby an abdominal stability program combined with gluteus medius training could strengthen the adductor muscles and improve lumbo-pelvic stability in female soccer players. 

## 2. Materials and Methods 

### 2.1. Study Design and Ethical Considerations

This randomized controlled study has been approved by the Research Committee of the European University of Madrid (registration number CIPI/18/090). After being informed of the purpose and experimental procedures of the study, a written informed consent was signed by all participants prior to the start of the study. The study was recorded in the International Clinical Trials Registry (ClinicalTrials.gov identifier: NCT03617887).

### 2.2. Study Population

The subjects were recruited from the female team of the C.F. Fuenlabrada soccer club. Samples were taken from January to April 2018. After contacting the club officials, the 25 soccer players were informed of the objectives and purposes of the study. All showed interest in participating in the study.

Of the 25 players that made up the sample at the beginning of the study, none dropped out during the intervention period or before the follow-up assessment. Therefore, all completed the study. Figure 1 shows the flowchart of the study. The study included only females with a mean age of 24.80 years (SD: 3.10) and a mean body mass index of 21.44 kg/m^2^ (SD: 2.70).

### 2.3. Selection Criteria

The inclusion criteria to participate in the study were: females; being aged between 18 and 35 years; currently federated with the Royal Football Federation of Madrid; players in the fifth level of competition of the Spanish female football league; and for at least 4 years; not having any type of pathology prior to or during the experimental period; and having signed the informed consent.

Exclusion criteria consisted of the following: past or current pathologies in any region of the lower limbs during the last 6 months; inability or failure to comply with the study requirements in terms of follow-up or involvement problems; female soccer players receiving other physiotherapy treatment or specific training at the time of the study; and soccer players under pharmacological treatment or undergoing physical therapy treatment parallel to the development of the study.

### 2.4. Randomization 

All the players were randomly allocated to the two study groups. Paper ballots with the letters IA represented the control group, and GM, the experimental group; these were placed in a ballot box. The subjects chose ballots, being assigned to one or the other study group: experimental (*n* = 13) or control (*n* = 12).

### 2.5. Sample Size 

A representative sample of the sample under study was calculated using G * Power software. The magnitude of this difference was considered by calculating the effect size (d = 2.2) [16], for the measurement of adduction strength in soccer players using a hand-held dynamometer. With an alpha level (type I error) of 0.05, a statistical power of 95% (1 − β = 0.95) and a non-sphericity correction of 1, a sample size of 9 female soccer players in each study group was estimated. Calculating an eventual 20% drop-out rate, the sample size should be 22 female soccer players.

### 2.6. Procedure

The intervention consisted of 20-min sessions, twice a week, for 8 weeks. The exercises consisted of abdominal training in the control group, and gluteus medius-specific exercises in addition to the above, in the experimental group. The exercise protocol [6,17,18,19] in common for both groups, included the performance of 3 exercises: Plank, Lateral plank and Bird dog (Figure 2).

Plank: Starting from the prone position, the soccer player kept her elbows aligned below her shoulders, forming a straight line perpendicular to the ground. The other point of support were the toes, raising the trunk and holding that position by means of an isometric contraction. Four 30-s repetitions were performed, with 30 s rest between repetitions.Lateral plank: Players were asked to lie in a lateral decubitus position supported at two points: forearm and feet, forming a “bridge” with the body. The contralateral arm stretches, in shoulder abduction, following the projection of the shoulder and the supporting arm. Six 10-s repetitions were performed, with 10 s rest between repetitions.Bird dog: From a quadrupedal position with the lumbar spine stabilized and back straight, players were asked to raise one arm and the contralateral leg, both parallel to the ground. They were asked to maintain stability and trunk control, causing abdominal muscle activation to prevent certain movements (pelvic scale or chest rotations). Three sets were performed, with 12 repetitions, each lasting 30 s [20].

The athletes included in the experimental group also performed two gluteus medius exercises: Pelvic drop and Stabilization of the gluteus medius in knee valgus (Figure 3).

Pelvic drop: From a standing position, with one foot on a box and the other on the floor, the player performed a pelvic tilt towards the same side. The supporting leg should be kept straight all the time and the abdominal muscles contracted, to counteract gluteus medius activation. It is important not to touch the floor with the downward foot, performing the exercise in a controlled manner. The position should be held for two seconds, then raising the pelvis and returning to the starting position, keeping the foot raised. Four sets were performed, 12 repetitions and each lasting 30 s [21,22,23,24].Stabilization of the gluteus medius in knee valgus: The player standing with a theraband at the height of the lower and lateral part of the knee to be treated, moves forward, keeping the supporting foot fixed in position. The theraband is held by the physiotherapist who exerts imbalance on the knee in the medial direction, while the player should hold that position for two seconds, limiting knee valgus by stabilizing the gluteus medius muscle. Five 12-s series were performed, with 30 s rest between sets [21,22,23,24,25].

### 2.7. Instruments 

Outcomes were assessed at baseline (T0), after intervention (T1), and at 4 weeks follow-up (T2). All subjects were assessed by the same rater, reproducing the same conditions and following the same measurement protocol. All players were evaluated by the same physiotherapist, blinded to subject allocation to study groups. Blinding of the evaluator was ensured by the evaluation procedure: upon arrival, the females were directed to an evaluation area, where the evaluator assessed all subjects at pre-treatment (T0), after the intervention (T1) and at follow-up (T2). The main outcome was lumbo-pelvic stability. The secondary outcome was adductor muscle strength.

The assessment of lumbo-pelvic stability was carried out using the protocol described by Cha et al. [26] with a universal goniometer. Position of the subject: supine decubitus with both legs raised forming a 90° angle and the lumbar area was stable and supported on the floor. Goniometer position: axis on the greater trochanter of the femur; fixed arm parallel to the floor; and mobile arm parallel to the lateral axis of the femur. From the starting position, the subject lowered her legs slightly and now when the lumbar spine lost contact with the floor, the angle of the legs relative to the floor was measured. The normal range is from 90° to 0° (0° being the optimal stability value). This measuring instrument has shown good test-retest reliability (ICC = 0.95; *p* < 0.05). 

Adductor muscle strength was measured according to the protocol developed by various authors [27,28] with a manual dynamometer. Position of the subject: standing. Position of dynamometer: attached to a fixed point, perpendicular and lateral to the leg to be treated, at the height of the distal third tibia. The subject performed hip adduction, generating a progressive maximum isometric contraction. The measurement was in Newtons.

### 2.8. Statistical Analysis

The statistical analysis was carried out with SPSS software, version 19.0, for Windows. The main statistics of the sample (mean and standard deviation) were obtained in all the sample variables and according to the group. The Shapiro-Wilk test was used to analyze the distribution of the sample in the two study groups. The parametric student’s *t*-test was used to obtain the differences in the dependent variables between the two groups. An analysis of variance of repeated measures was carried out to compare the two groups (experimental and control) at the three assessment times: baseline (T0), posttreatment (T1) and follow-up (T2). To control the error rate of the significance level, the Bonferroni correction method was applied. The results of the F test depend on the significance of Mauchly’s sphericity test. If significant, the Greenhouse–Geisser correction was used. The effect size of changes for all the variables analyzed after the training intervention was calculated as partial eta square (ηp^2^) from the ANOVAs (small: 0.01; medium: 0.06; and large: 0.14) [29]. For subjects dropping out, an intention to treat analysis would be carried out. In all statistical tests the significance level is α = 0.05.

## 3. Results

At baseline, there were only significant differences between the two groups regarding the variables for body mass index (*p* = 0.04) and competition time played in the current season (*p* = 0.03). Similarly, no differences were detected between groups for any of the dependent variables at baseline (see Table 1). The main dispersion measures (mean and standard deviation) of the dependent variables were calculated in each of the evaluations (see Table 2).

### 3.1. Primary Outcome

After the intervention period significant changes were only found in lumbo-pelvic stability in the control group (*p* < 0.01; CI95 = 3.11–16.04). However, changes in the experimental group were not significant (*p* = 0.08; IC95 = −0.66–10.36). When comparing pre-treatment and follow-up evaluations, significant changes were only noted in the control group (*p* = 0.02; IC95 = 0.92–9.73) (see Table 3).

Significant differences were shown in the repeated measures factor depending on the moment evaluated in lumbo-pelvic stability (F (1.19, 27.58) = 12.41; *p* < 0.01; η^2^_p_ = 0.35), but not in group interaction (F = 1.32; *p* = 0.26; η^2^_p_ = 0.05) (see Table 4). 

The test for equality of covariance matrices disclosed no interaction between the type of intervention and the response in the evaluations (*p* = 0.87). When performing the pairwise comparison analysis, we found significant differences between pre-treatment and post-treatment evaluations (*p* < 0.01), between pre-treatment and follow-up evaluations (*p* = 0.02), and between post-treatment and follow-up evaluations (*p* < 0.01) (see Table 5).

### 3.2. Secondary Outcome

Following intervention, we found significant changes in the experimental group in the left adductor (*p* < 0.001; IC95 = −3.43–−1.53) and right (*p* = 0.02; IC95 = −2.76–−0.19) muscle strength. The control group also revealed differences after the intervention in left (*p* < 0.001; IC95 = −2.30–−1.07) and right (*p* < 0.001; IC95 = −2.77–−1.33) adductor strength. When comparing pretreatment and follow-up evaluations, we found significant differences in the experimental group in left adductor strength (*p* < 0.01; IC95 = −2.61–−0.59) and in the control group in right (*p* < 0.001; IC95 = −2.26–−0.99) and left (*p* < 0.01; IC95 = −2.00–−0.75) adductor strength. 

In the repeated measures factor, we found differences depending on the moment evaluated in left (F (1.21, 28.03) = 44.22; *p* < 0.001; η2p = 0.65) and right (F (1.19, 27.49) = 23.64; *p* < 0.001; η2p = 0.50) adductor strength. However, not so in group interaction for left (F = 1.62; *p* = 0.21; η^2^_p_ = 0.06) and right (F = 0.94; *p* = 0.35; η^2^_p_ = 0.04) adductor strength. The test of the equality of covariance matrices revealed interaction between the type of intervention and the response in the evaluations of the right adductor strength variable (*p* = 0.01); not, however, for the left adductor (*p* = 0.70). 

With regard to the pairwise comparison analysis, in the evaluation of the left and right adductor strength, there were significant differences between the pre- and post-treatment evaluations (*p* < 0.001), between pre-treatment and follow-up evaluations (*p* < 0.001), and between post-treatment and follow-up evaluations (*p* < 0.01).

## 4. Discussion

The aim of our study was to test the effectiveness of an abdominal training program and gluteus medius-specific training for improving lumbo-pelvic stability and adductor muscle strength in female soccer players. Both interventions improved lumbo-pelvic stability and adductor muscle strength to the same extent, without evidence of one being more effective than the other. Although the sample size was small, effect size values were high in the intragroup effect analysis. The absence of significant differences in group interaction suggested that the improvement in the variables was not dependent on the group allocation of the soccer players. Thus, we can establish that both interventions were effective in improving lumbo-pelvic stability and adductor muscle strength in the soccer players recruited in this study.

Few scientific articles have been found in the literature that provide data on the incidence, etiology, prevention or treatment of injuries in female’s soccer. Rosas et al. [30] conducted a study with the largest sample size (*n* = 25), equivalent to that recruited in the present study. Although intervention programs have been developed based on neuromuscular, plyometric, strength or power improvement training [31,32,33], there are discrepancies as to which exercises are the most effective in improving the performance of female soccer players. A recent review [34] has reported that plyometric training is more effective for improving vertical jump performance, linear sprint, and change of direction as compared to strength training in soccer players. 

Our study has been developed based on different work protocols in healthy subjects, focusing on the increase of stability and strength, the ultimate goal being to establish the potential impact on the prevention of injuries. The most comprehensive protocols in terms of muscle groups, are those developed by Hölmich et al. [8] and Krommes et al. [9] applied to the adductor muscles, and by Krause et al. [22], on activation of the gluteus medius. To date, no article has developed an intervention program addressing more than one muscle group or different structures. Our study aims to assess the efficacy of an exercise program which targets different muscle groups and structures.

Grygorowicz et al. [35] noted that, depending on the intensity of the training, there are alterations in movement patterns, whereby prevention programs need to be adapted to the soccer player’s level of skill. However, female soccer players who only performed abdominal training (control group), improved the values related to adductor muscle strength and lumbar-pelvic stability. It should be noted that the protocols developed by Hölmich et al. [8] and Krommes et al. [9] only accounted for the adductor strength variable. Based on our results, abdominal training may have an impact on neighboring structures.

On the other hand, the experimental group exhibited increased improvement of adductor strength and lumbar-pelvic stability. Published studies [22,23,25] on gluteus medius training alone have not shown improvements when implemented through a variety of exercises. However, Monteiro et al. [21] reported gluteus medius activation achieved through a protocol based on a single exercise (pelvic drop). This may suggest that excessive activation of this muscle could lead to fatigue and, hence, overtraining can be counterproductive.

Synergistic control between the pelvis, spine, and trunk involves dynamic and static postures that influence performance results for low- and high-level tasks. Athletes often have positive self-perceptions associated with athletic performance [36]. Changes in self-perceptions are positively correlated with improvements in strength and fitness after a training intervention [37,38]. Our results may contribute to promote the development of an abdominal training protocol with the long-term goal of providing benefits in terms of injury prevention, without fatiguing the gluteus medius muscle. This would offer the advantage of saving time in training as the injury prevention protocol would not focus on the gluteus medius. 

A noteworthy aspect is that, despite not finding intergroup differences, the improvement in lumbo-pelvic stability and adductor muscle strength was maintained after a 4-month follow-up period in both groups. The practical application of both protocols could be useful in the planning of soccer schedules for a specific competitive period with the purpose of improving sports performance. 

### 4.1. Study Limitations

This study presents some limitations that could affect the interpretation of the results supporting the hypothesis, and this should be taken into account in future studies to overcome these difficulties. The low sample size is the main limitation of this study, so the results need to be interpreted with caution. A larger number of methodological quality contributions (blinding of the rater, calculation of the interobserver reliability analysis, follow-up assessment, etc.) have been implemented to compensate for the small simple size, thus designing a study with acceptable methodological quality. 

There are also psychological, physiological or nutritional factors that can affect the sports performance of each player, these being beyond our control, and which might have altered their performance, directly affecting the results of the study. 

Following the intervention, no differences were found between the two study groups. The implementation of gluteus medius specific training combined with the abdominal program failed to provide additional gains for adductor strength and lumbo-pelvic stability. These results were not part of the initial hypothesis of this study. A longer training period or an adaptation of the exercises of the intervention protocol could provide a greater improvements compared to the abdominal program alone. Future studies should address this possibility.

### 4.2. Relevance to Clinical Practice

The results obtained may reinforce some of the concepts already existing in the area of injury prevention in soccer. In the first place, an attempt is made to establish the relationship, in terms of muscle and myofascial chains, between the abdominal and lumbar-pelvic central stability region, and the adductor muscles. This is one of the structures most often injured in soccer players, a possible cause often being poor stability of the central core itself.

The results of this study can serve as the basis for prevention plans in female soccer players, as the incidence of injury, although not the same, is similar. In the same way, the injured structures tend to be the same so it can be used in both sexes. Moreover, the plan described would involve a low-cost resource with short sessions, something which favors its applicability.

Lastly, the aim is to promote and contribute to the prevention of diseases in women’s soccer, and to provide evidence towards the prevention of diseases that currently account for a significant proportion of injuries in soccer. 

### 4.3. Recommendations for Future Research

Future studies would ideally involve a longer research period, to deepen in the search of relevant scientific articles and in the design of an intervention plan suited to the needs of the population. A larger sample size is essential to improve the strength of the results. Lastly, this growing line of research in the field of women’s soccer should be encouraged, and more specifically in the field of injury prevention, since this represents an important factor to increase and optimize sports performance.

## 5. Conclusions

This study evidenced how eight weeks of abdominal and gluteus medius stability, as well as strength training, can improve lumbo-pelvic stability and adductor strength in female footballers. Gluteus medius-specific training in addition to the abdominal training program fails to provide additional gains for adductor strength and lumbo-pelvic stability. The improvements in adductor strength and stability in both groups were maintained after a 4-week follow-up period. 

## Figures and Tables

**Figure 1 ijerph-18-01528-f001:**
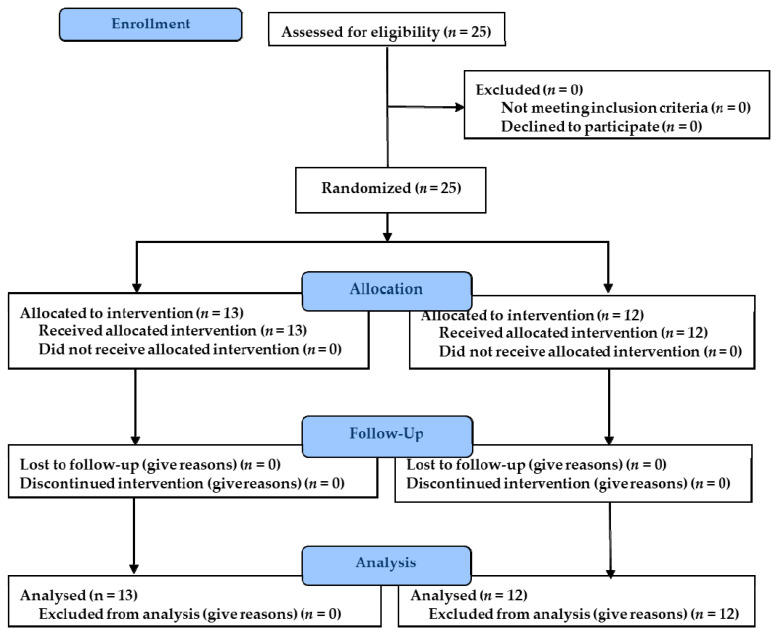
Flowchart of the study.

**Figure 2 ijerph-18-01528-f002:**
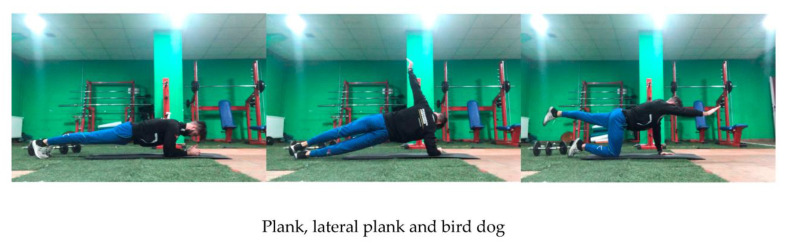
Show the three exercises.

**Figure 3 ijerph-18-01528-f003:**
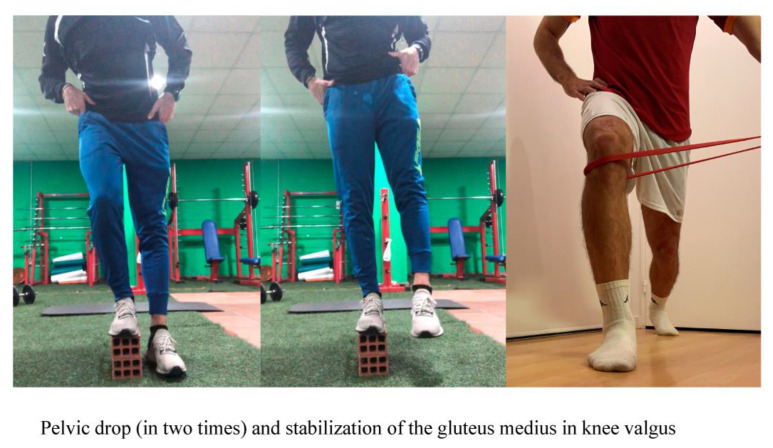
Show both exercises.

**Table 1 ijerph-18-01528-t001:** Baseline characteristics (median and interquartile range) for the sample and depending on the study group.

Variables	Experimental Group	Control Group	*p*-Value ^a^
Age (years)	23.92 (2.84)	25.75 (3.22)	0.30
Body mass index (Kg/m^2^) *	21.56 (3.19)	21.32 (2.19)	0.04
Time played (minutes) *	2060.77 (288.13)	1122.50 (750.22)	0.03
Lumbar-pelvic stability	59.46 (6.02)	63.33 (5.39)	0.24
Left leg strength	15.27 (1.85)	16.75 (1.76)	0.41
Right leg strength	16.70 (2.39)	16.46 (1.50)	0.38

^a^ Shapiro–Wilks test; * Significant difference (*p* < 0.05).

**Table 2 ijerph-18-01528-t002:** Descriptive statistics (mean and standard deviation) of the study variables at baseline, posttreatment and follow-up assessments.

Variables	Experimental Group			Control Group		
	T0	T1	T2	T0	T1	T2
Lumbar-pelvic stability ^║^	59.46 (6.02)	54.61 (9.97)	56.23 (7.62)	63.33 (5.39)	53.75 (8.40)	58.00 (6.16)
Left leg strength ^‡^	15.27 (1.85)	17.76 (1.83)	16.88 (1.83)	16.75 (1.76)	18.44 (1.76)	18.13 (1.53)
Right leg strength ^‡^	16.70 (2.39)	18.18 (2.92)	17.66 (2.65)	16.46 (1.50)	18.52 (1.41)	18.09 (1.31)

Outcome measures at the baseline (T0), after the three-week period of experimental and control interventions (T1) and after further 4-weeks as follow-up (T2). ^‡^ Higher scores indicating greater strength; ^║^ 90 to 0 degrees, with higher scores indicating less stability.

**Table 3 ijerph-18-01528-t003:** Means difference and significance between the three evaluations carried between T0–T1 and T0–T2 assessments.

Variables	Experimental Group		Control Group	
	T0–T1	T0–T2	T0–T1	T0–T2
Lumbar-pelvic stability (degree)	4.84	3.23	9.58 *	5.33 *
Left leg strength (Newtons)	−2.48 **	−1.60 *	−1.68 **	−1.38 **
Right leg strength (Newtons)	−1.48 *	−0.95	−2.05 **	−1.63 **

T0–T1: outcome measures between baseline to posttreatment assessments: T0–T2: outcome measures between baseline to follow-up assessments. * Significant difference between improvements of the study groups (*p* < 0.01). ** Significant difference between improvements of the study groups (*p* < 0.001).

**Table 4 ijerph-18-01528-t004:** Statistical analysis of repeated measures of the dependent variables in the baseline, posttreatment and follow-up assessments.

Variables	Mauchly Sphericity Test		Intra-Subject Effect			Intra-Subject Effect		
	W	Sig.	F	Sig.	η^2^_p_	F	Sig.	η^2^_p_
Lumbar-pelvic stability (degree) ^a^	0.33	0.00	12.41	0.00	0.35	1.32	0.26	0.05
Left leg strength (Newtons) ^a^	0.35	0.00	44.22	0.00	0.65	1.62	0.20	0.06
Right leg strength (Newtons) ^a^	0.32	0.00	23.64	0.00	0.50	0.94	0.35	0.04

W: Mauchly Sphericity Test; Sig.: significance; F: dispersion statistic; η^2^_p_: eta squared partial. ^a^ The df corresponds to Greenhouse–Geisser test.

**Table 5 ijerph-18-01528-t005:** Pairwise comparison analysis (means difference and (significance)) between the three evaluations carried out in each study group.

Variables	T0–T1	T1–T2	T0–T2
Lumbar-pelvic stability (degree)	7.21 (0.00) **	−2.93 (0.00) **	4.28 (0.01) *
Left leg strength (Newtons)	−2.08 (0.00) **	0.59 (0.00) **	−1.49 (0.00) **
Right leg strength (Newtons)	−1.76 (0.00) **	0.47 (0.00) **	−1.29 (0.00) **

T0–T1: outcome measures between baseline to posttreatment assessments; T1–T2: outcome measures between posttreatment to follow-up assessments: T0–T2: outcome measures between baseline to follow-up assessments (T0); MD, mean difference. * Significant difference between improvements of the study groups (*p* < 0.05) ** Significant difference between improvements of the study groups (*p* < 0.001).

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
