# Peer review of "Effectiveness of Abdominal and Gluteus Medius Training in Lumbo-Pelvic Stability and Adductor Strength in Female Soccer Players. A Randomized Controlled Study"

_ijerph, 2021, doi:10.3390/ijerph18041528_

Round 1

Reviewer 1 Report

Overall, the manuscript was interesting. I suggest a revision on some of the English wording throughout to make it a little cleared to read and more concise. The general understanding is there, but the manuscript needs a little work. 

Some images to demonstrate the exercises in the methods section would be ideal so please add these if possible.

There was no mention of whether the subjects were performing external gym or exercise outside of the investigation. If they were undertaking other rehabilitation or strength and conditioning work, this is very likely to have influenced the results so this needs to be at least mentioned as a limitation.

Some more discussion would be good as this section is very brief. Perhaps a little more discussion about previous research which has lead to this one? Maybe some more discussion about the gaps in the research eg. nothing has been found on this previously so how is this so? What papers/research has demonstrated no instrumental findings?

Author Response

Reviewer 1

Comments and Suggestions for Authors

Overall, the manuscript was interesting. I suggest a revision on some of the English wording throughout to make it a little cleared to read and more concise. The general understanding is there, but the manuscript needs a little work.

  • Some images to demonstrate the exercises in the methods section would be ideal so please add these if possible. We have included figures to help the reader understand the intervention carried out.
  • There was no mention of whether the subjects were performing external gym or exercise outside of the investigation. If they were undertaking other rehabilitation or strength and conditioning work, this is very likely to have influenced the results so this needs to be at least mentioned as a limitation. None of the athletes was undergoing physiotherapy treatment or with a specific treatment. To avoid confusion for the reader, we have included it as a selection criterion.
  • Some more discussion would be good as this section is very brief. Perhaps a little more discussion about previous research which has lead to this one? Maybe some more discussion about the gaps in the research eg. nothing has been found on this previously so how is this so? What papers/research has demonstrated no instrumental findings? We have included sections in the Discussion with the results of previous studies, comparing the results with those observed in our study.

Reviewer 2 Report

I would like to thank for your effort to carry out this research. I will send some comments with the intention to improve the manuscript

Abstract: 

Line 16. "randomly..." lack of a verb

Line 21. What "surgery"?

Line 21. Explain Acronym

Line 28. Which improvements? In which groups?

Introduction:

Line 39-42: Example of too long phrases in the text. An overall review of english would improve the manuscript

Material and methods

Line 78: m2 (2 with superscript)

Figure 1: 12 can not be excluded from analysis (lower right part of the figure)

Line 82: include female and level of sport performance (type of league) in the selection criteria

Line 88: any kind of physical therapy as exclusion criteria?

Sample size: Sample size is determined with variables that are not studied in this manuscript. Please Explain.

Line 119: Are these two exercises really specific for the gluteus medius? I do not have a problema with the description but a comment about real specificity could be addressed in discussion.

Line 167: standardize verb in past throughout methods section

Results

Tables. Detail acronyms in the legends of the tables.

Discussion

Line 246. Normal sporting and training activity could be also explaining the improvement in adductor muscle strength.

The purpose of this study was to compare the efficacy of an abdominal and gluteus medius training protocol in terms of improving lumbar-pelvic stability and strength of the adductor muscles, compared to abdominal training alone, in semi-professional female soccer players aged 58 from 18 to 35 years.
It is not enough developed in the discussion.

Line 264. These quality contributions do not compensate the low sample size.

Line 275. Both sexes? In the introduction, the focus was to make a specific approach for females. It seems contradictory.

Lines 279-282. Not analyzed in this study.

Conclusions

Line 293 and lines 296-297: some conclusions are not related to the objective of the study neither being analyzed in the study

Author Response

Reviewer 2

Comments and Suggestions for Authors

I would like to thank for your effort to carry out this research. I will send some comments with the intention to improve the manuscript

Abstract:

  • Line 16. "randomly..." lack of a verb. After revision of the text, were has been added before the word randomly.
  • Line 21. What "surgery"? We have added intervention. Due to an error in the translation, the word surgery was wrongly included.
  • Line 21. Explain Acronym. The meaning of MD has been included.
  • Line 28. Which improvements? In which groups? The improvements that were maintained after the follow-up period occurred in the two study variables. We have included this aspect in the sentence to avoid doubts to the reader.

Introduction:

  • Line 39-42: Example of too long phrases in the text. An overall review of english would improve the manuscript. A professional translator has reviewed the entire text as a whole, according to the comments of the three reviewers.

Material and methods

  • Line 78: m2 (2 with superscript). Has been set correctly.
  • Figure 1: 12 can not be excluded from analysis (lower right part of the figure). There was indeed an error in the flow chart. We have corrected it.
  • Line 82: include female and level of sport performance (type of league) in the selection criteria. The two inclusion criteria indicated have been included.
  • Line 88: any kind of physical therapy as exclusion criteria? Although it was a specific exclusion criterion, it had not been included. It has already been indicated in the text.
  • Sample size: Sample size is determined with variables that are not studied in this manuscript. Please Explain. We include the reference of a parallel article, and by mistake we include it. We apologize for the error. We have updated the article that was taken as a reference, where the adductor strength in soccer players was evaluated.
  • Line 119: Are these two exercises really specific for the gluteus medius? I do not have a problema with the description but a comment about real specificity could be addressed in discussion. As the reviewer points out, there may be some debate regarding the specificity of gluteus medius exercises. For this reason, and to avoid confusion for the reader, we have eliminated the word specific.
  • Line 167: standardize verb in past throughout methods section. A professional translator has reviewed the entire text as a whole, according to the comments of the three reviewers.

Results

  • Detail acronyms in the legends of the tables. We have included the description of all the acronyms of the tables.

Discussion

  • Line 246. Normal sporting and training activity could be also explaining the improvement in adductor muscle strength. This option was not even considered since one of the exclusion criteria was to receive another intervention or physiotherapy treatment. Similarly, when continuing with their physical activity of training and competition, as in the previous period of the study, there was no other factor that could influence the results beyond the intervention applied in this study.
  • The purpose of this study was to compare the efficacy of an abdominal and gluteus medius training protocol in terms of improving lumbar-pelvic stability and strength of the adductor muscles, compared to abdominal training alone, in semi-professional female soccer players aged 58 from 18 to 35 years. It is not enough developed in the discussion. We have included in the Discussion section a greater comparison of results and previous studies to facilitate the understanding of the text.
  • Line 264. These quality contributions do not compensate the low sample size. Indeed, as indicated by the reviewer, the methodological quality measures cannot compensate for the sample size. we have redrafted the sentence to facilitate understanding and not to leave room for doubt.
  • Line 275. Both sexes? In the introduction, the focus was to make a specific approach for females. It seems contradictory. As the reviewer points out, this rase can lead to confusion. We have included the reference to female soccer players, which is the correct phrase.
  • Lines 279-282. Not analyzed in this study. To avoid confusion for the reader, we have removed that phrase.

Conclusions

  • Line 293 and lines 296-297: some conclusions are not related to the objective of the study neither being analyzed in the study. As the reviewer points out, we have modified the Conclusions section to avoid confusion, sticking to the results of the study.

Reviewer 3 Report

Thank you for the opportunity to review your paper, which considers some interesting, applied issues for the field of female soccer injury prevention. Female soccer players suffer from specific sport related injuries, where the application of evidence-based neuromuscular prevention programs is essential to reduce the likelihood of these lower-limb injuries. In this sense, this is a timely issue that explores the efficacy of a training program on lumbo-pelvic stability of female soccer players. The methods and results are clearly exposed and the controlled desing and blindness add strength to the paper. However, there are some aspects I would like to comment in order to improve the readability and understanding of the study especially in regards to the introduction.

General Comments:

  • The grammar could improve substantially to the ease readability of the manuscript. I suggest a review from an English native speaker.
  • I would like to recommend the addition of an illustrative figure with the exercises performed by the athletes because some journal readers may not have a clear understanding of this type of exercise or execution form.
  • Consider using the term female soccer players instead of women for the title and then maintain the same terminology for the rest of the paper.
  • The design and study population is intended to female soccer players. I suggest the introduction could be more hypotheses driven where the authors address the specific epidemiology of lower-limb injuries and specific adductor or lumbo-pelvic injuries in female soccer players and the rationale behind the importance of adding specific exercises targeting the lumbo-pelvic muscles and the gluteus medius to address abdominal and lower-limb imbalances. For example, the reader may have some questions regarding why the authors added gluteus medius exercises? Why are these so important for female soccer players?
  • Review all references. For example in line 143 Cha et al (17) but in the reference list 17 is Ramirez-Campillo et al.

Specific comments:

Abstract

  • Line 21: “lumbo-pelvic stability after surgery”. Consider deleting after surgery because it seems to be not part of the paper.
  • Line 16: poor readability of this phrase. Consider re-writing to “were randomly..”
  • Line 21-23: consider re-writing for better clarity of results.
  • Line 28: the authors mention flexibility as an outcome measure in the conclusions of the abstract. However, it was not mentioned as an outcome measure in the methods. Consider adding or deleting adductor flexibility in the conclusion.
  • Line 29. Similar to the previous point, the authors mention a 4-week follow-up. However, this was not mentioned in the methods. Consider adding to the methods the 3 assessment periods.

Introduction

  • Line 47..”important programs” of what? Injury prevention …consider clarifying what type of program was proposed by Holmich et al.
  • Line 54… consider changing the term “operation”. It is unclear what the authors are explaining in this line.
  • Line 56, The purpose of the study seems to be to test the hypothesis that adding targeted gluteus medius exercises to the abdominal training program is more effective in improving lumbo-pelvic stability and the adductor muscle’s strength. Consider re-writing this important part of the paper for better readability.
  • Line 60. “abdominal stability and strength training program”. Did the program consist of strength training+ abdominal + gluteus? Maybe consider deleting strength training .
  • Line 61. Add “and improve lumbo-pelvic stability”.

Methods.

  • Line 142. Lumbo-pelvic stability test. Could the authors add some references regarding the test’s reliability-validity in the literature?
  • Maybe consider inserting table 1 and 2 around line 173 so the tables seem more disperse throughout the result text.

Discussion

It seems worthwile to discuss what the authors think or hypothesize regarding the results of the control group on adductor strength. It seems that already doing the core exercises there are almost the same results? In the discussion the authors summarize the results “ both interventions improved the outcome measures to the same extent with no evidence of one being more effective than the other”. In other words, no differences between groups were found. Therefore, adding glutes medius to the abdominal program did not provide additional gains to adductor strength and core stability. So, the results were not expected from the initial hypothesis. Maybe in limitations the authors could discuss what could they have changed in the protocol to have demonstrated their hypothesis? Longer period of training, different protocol?  

Line 234. Add female soccer players.

Line 245. consider changing to “Different muscle groups” rather than tissues.

Line 283. Change diseases to injury.

Also, add a discussion and conclusion phrase regarding the follow-up results achieved. 

Author Response

Reviewer 3

Comments and Suggestions for Authors

Thank you for the opportunity to review your paper, which considers some interesting, applied issues for the field of female soccer injury prevention. Female soccer players suffer from specific sport related injuries, where the application of evidence-based neuromuscular prevention programs is essential to reduce the likelihood of these lower-limb injuries. In this sense, this is a timely issue that explores the efficacy of a training program on lumbo-pelvic stability of female soccer players. The methods and results are clearly exposed and the controlled desing and blindness add strength to the paper. However, there are some aspects I would like to comment in order to improve the readability and understanding of the study especially in regards to the introduction.

General Comments:

  • The grammar could improve substantially to the ease readability of the manuscript. I suggest a review from an English native speaker. The text has been checked by a professional translator.
  • I would like to recommend the addition of an illustrative figure with the exercises performed by the athletes because some journal readers may not have a clear understanding of this type of exercise or execution form. We have included figures to help the reader understand the intervention carried out.
  • Consider using the term female soccer players instead of women for the title and then maintain the same terminology for the rest of the paper. The word women has been changed to female throughout the text.
  • The design and study population is intended to female soccer players. I suggest the introduction could be more hypotheses driven where the authors address the specific epidemiology of lower-limb injuries and specific adductor or lumbo-pelvic injuries in female soccer players and the rationale behind the importance of adding specific exercises targeting the lumbo-pelvic muscles and the gluteus medius to address abdominal and lower-limb imbalances. For example, the reader may have some questions regarding why the authors added gluteus medius exercises? Why are these so important for female soccer players? Based on the comments of two of the reviewers, we have made changes to the Introduction section to facilitate understanding of the background of the study topic.
  • Review all references. For example in line 143 Cha et al (17) but in the reference list 17 is Ramirez-Campillo et al. We have updated all references to avoid errors like that.

Specific comments:

Abstract

  • Line 21: “lumbo-pelvic stability after surgery”. Consider deleting after surgery because it seems to be not part of the paper. We have added intervention. Due to an error in the translation, the word surgery was wrongly included.
  • Line 16: poor readability of this phrase. Consider re-writing to “were randomly..” After revision of the text, were has been added before the word randomly.
  • Line 21-23: consider re-writing for better clarity of results. We have redrafted the phrase to facilitate your understanding.
  • Line 28: the authors mention flexibility as an outcome measure in the conclusions of the abstract. However, it was not mentioned as an outcome measure in the methods. Consider adding or deleting adductor flexibility in the conclusion. Due to an error in the writing of results, this term was erroneously included. It has been fixed to avoid confusion.
  • Line 29. Similar to the previous point, the authors mention a 4-week follow-up. However, this was not mentioned in the methods. Consider adding to the methods the 3 assessment periods. The 3 study evaluation periods have been included in the abstract.

Introduction

  • Line 47..”important programs” of what? Injury prevention …consider clarifying what type of program was proposed by Holmich et al. The sentence has been redrafted to make it easier to understand.
  • Line 54… consider changing the term “operation”. It is unclear what the authors are explaining in this line. We have changed the phrase to facilitate your understanding.
  • Line 56, The purpose of the study seems to be to test the hypothesis that adding targeted gluteus medius exercises to the abdominal training program is more effective in improving lumbo-pelvic stability and the adductor muscle’s strength. Consider re-writing this important part of the paper for better readability. We have changed the phrase to facilitate your understanding
  • Line 60. “abdominal stability and strength training program”. Did the program consist of strength training+ abdominal + gluteus? Maybe consider deleting strength training. We have removed the term strength training, as indicated by the reviewer.
  • Line 61. Add “and improve lumbo-pelvic stability”. As the reviewer points out, we have added the term improve.

Methods.

  • Line 142. Lumbo-pelvic stability test. Could the authors add some references regarding the test’s reliability-validity in the literature? As indicated by the reviewer, we have included the reliability values of Cha et al. (2017).
  • Maybe consider inserting table 1 and 2 around line 173 so the tables seem more disperse throughout the result text. We have made the change indicated by the reviewer.

Discussion

  • It seems worthwile to discuss what the authors think or hypothesize regarding the results of the control group on adductor strength. It seems that already doing the core exercises there are almost the same results? In the discussion the authors summarize the results “ both interventions improved the outcome measures to the same extent with no evidence of one being more effective than the other”. In other words, no differences between groups were found. Therefore, adding glutes medius to the abdominal program did not provide additional gains to adductor strength and core stability. So, the results were not expected from the initial hypothesis. Maybe in limitations the authors could discuss what could they have changed in the protocol to have demonstrated their hypothesis? Longer period of training, different protocol? Indeed, the results do not agree with the initial working hypothesis. As indicated by the reviewer, we have included this aspect in the Study Limitations
  • Line 234. Add female soccer players. We have added the word female.
  • Line 245. consider changing to “Different muscle groups” rather than tissues. We have changed the word, as the reviewer points out.
  • Line 283. Change diseases to injury. We have made the indicated change.
  • Also, add a discussion and conclusion phrase regarding the follow-up results achieved. We have added a reference to follow-up results and effect size in the Discussion Similarly, we have added a sentence in the Conclusions section.

Round 2

Reviewer 2 Report

Thanks for the work adapting the paper